# Edge-free but Structure-aware: Prototype-Guided Knowledge Distillation from GNNs to MLPs

## Abstract

Distilling high-accuracy Graph Neural Networks (GNNs) to low-latency multilayer perceptrons (MLPs) on graph tasks has become a hot research topic. However, conventional MLPs almost exclusively rely on the graph nodes and fail to effectively capture the graph structural information. Previous methods address this issue by processing graph edges into extra inputs for MLPs, but such graph structures may be unavailable for various scenarios. To this end, we propose Prototype-Guided Knowledge Distillation (PGKD), which does not require graph edges (edge-free setting) yet learns structure-aware MLPs. Our insight is to distill graph structural information from GNNs. Specifically, we first analyze the impact of graph structures on GNN teachers, and then design two losses based on prototypes to distill such information from GNNs to MLPs. Experimental results on popular graph benchmarks demonstrate the effectiveness and robustness of the proposed PGKD.

## 1    Introduction

Graph Neural Networks (GNNs) are gaining importance in handling non-Euclidean structural data and have achieved start-of-the-art performance across graph machine learning tasks, particularly for the node classification task (Kipf & Welling, 2017; Velickovic et al., 2017; Hamilton et al., 2017). The message-passing architecture, which aggregates the information from neighborhoods, guarantees the powerful representation ability in GNNs. However, such neighborhood fetching operation also leads to high latency (Jia et al., 2020), making it challenging to apply GNNs for real-world applications. Meanwhile, MLPs are free from the GNN latency problem without message-passing architecture, but perform poorly in graph tasks due to the lack of graph structural information. Therefore, it is challenging to train low-latency MLPs to have competitive accuracy as GNNs on graph tasks.

To achieve this goal, one mainstream approach is to distill the knowledge from GNNs to MLPs. GLNN (Zhang et al., 2022) employs the vanilla logit-based knowledge distillation (KD) to train MLP students from GNN teachers. Despite the KD target, the MLPs in GLNN still suffer from the lacking of graph structural information, since MLPs rely exclusively on the node features. To inject the graph structural information, previous methods (Hu et al., 2021; Wu et al., 2022) employ additional regularization terms on the final node representations based on the graph structures. For each node, the key insight is to draw closer the distance between the selected node and its $i$-hop connected neighbors while pushing further for other nodes. However, this strategy has **two issues**: 1) Graph edges are required as an auxiliary input, but graph structures may be **unavailable** for MLP students for various reasons, including privacy problems, commercial considerations, and missing/corrupted edges (please refer to A.2 for details); 2) Such prior knowledge, which the regularization terms rely on, is **irrelevant** to the GNN teachers. Therefore, how to distill the GNN teachers to structure-aware MLP students while graph edges are unavailable (viz. an edge-free setting) becomes an important topic. We thus ask: *What is the impact of graph structures (i.e. graph edges) on GNNs? Can we distill such graph structural information from GNNs to MLPs so that we can get structure-aware MLPs in an edge-free setting?*

To answer this, we first analyze the impact of graph structures on GNNs. We categorize the graph edges into **intra-class** and **inter-class** edges, where the nodes connected by the edge are from the same

Table 1: Comparison among different methods to distill GNNs to MLPs. **Edge-free** denotes whether graph edges are employed as extra inputs during distillation. **Structure-aware** denotes whether the learned MLPs are aware of the graph structural information.

| Methods | Edge-free | Structure-aware |
|---|---|---|
| Vanilla KD (GLNN (Zhang et al., 2022)) | ✓ | ✗ |
| Regularization-based (Hu et al. (2021); Wu et al. (2022)) | ✗ | ✓ |
| PGKD (ours) | ✓ | ✓ |

and different classes, respectively. The intra-class edges enforce the local smoothness by constraining the learned representations of two connected nodes to become similar, so that the homophily property for nodes from the same class can be captured (Zhu et al., 2021). The inter-class edges, connecting two nodes from different classes, determine the pattern of distances among these classes.

Based on the analysis, we propose Prototype-Guided Knowledge Distillation (PGKD), employing the class prototype (viz. a typical embedding vector of a given class (Snell et al., 2017)) for distillation. Specifically, we design extra alignment losses for MLPs based on the class prototypes, aiming to mimic the impact of graph structures (i.e. graph edges) on GNNs. In PGKD, we first design a prototype strategy to get all class prototypes for both GNN teachers and MLP students. To mimic the intra-class edges, we force the node representations from MLPs closer to their corresponding prototypes. Meanwhile, we align the MLP prototypes with GNN prototypes to learn the inter-class distance patterns. As shown in Table 1, PGKD is the **first** method to distill GNNs to structure-aware MLPs in an edge-free setting.

We perform experiments on popular graph benchmarks in both transductive and inductive settings. We also conduct extensive ablation studies and analyses on PGKD. Empirical results demonstrate the effectiveness and robustness of PKGD. In short, our main contributions are: **1)** We analyze the impact of graph structures on GNNs by categorizing the edges into intra-class edges and inter-class edges, thus providing a deeper understanding of GNNs. **2)** We propose PGKD, the *first-ever* method to distill GNN teachers to structure-aware MLP students in the edge-free setting. **3)** We evaluate PGKD on various graph benchmarks and demonstrate its effectiveness and robustness.

## 2 RELATED WORK

### 2.1 DISTILLING GNNS TO MLPS

Knowledge Distillation (KD) is among the mainstream approaches to transferring knowledge from GNNs to MLPs. The key insight is to learn a student model by mimicking the behaviors of the teacher model. GLNN (Zhang et al., 2022) utilizes the vanilla logit-based KD (Hinton et al., 2015), which is edge-free but fails to capture the graph structural information. To address this issue, one way is to process graph edges into extra inputs for MLPs, such as the adjacency matrix (Chen et al., 2022) or the node positions (Tian et al., 2022). Another line is to treat graph structural information as a regularization term, where the nodes connected by edges should be closer (Wu et al., 2022; Hu et al., 2021). Nonetheless, graph structure may be unavailable for some reasons, including privacy problems, commercial considerations, and missing/corrupted edges. In this work, we propose PGKD, the first method to distill GNNs to structure-aware MLPs without graph structure.

### 2.2 PROTOTYPE IN GNNS

Prototypical Networks (Snell et al., 2017) have been widely applied in few-shot learning and metric learning on classification tasks (Huang & Zitnik, 2020). The basic idea is that there exists an embedding in which points cluster around a single prototype representation for each class. In GNNs, class prototypes are widely employed for few-shot classification (Satorras & Estrach, 2018; Yao et al., 2020), zero-shot classification (Wang et al., 2021), graph matching (Wang et al., 2020), and graph explanation (Shin et al., 2022; Ying et al., 2019). The class prototypes are usually defined as simple as mean vectors. In this work, we design extra losses for MLP students via prototypes, aiming to distill the graph structural information from GNN teachers. To the best of our knowledge, this is the *first-time* utilization of prototypes for GNN-to-MLP distillation.

## 3 PRELIMINARIES

**Notations.** Let $\mathcal{G} = (\mathcal{V}, \mathcal{E})$ denote a graph, where $\mathcal{V}$ stands for all $N$ nodes with features $\mathbf{X} \in \mathbb{R}^{N \times D}$ and $\mathcal{E}$ stands for all edges. We represent edges with an adjacency matrix $\mathbf{A}$, and $A_{u,v} = 1$ if edge $(u, v) \in \mathcal{E}$ or 0 otherwise. For the node classification task, the target is $\mathbf{Y} \in \mathbb{R}^{N \times K}$, where row $y_v \in R^K$ denotes the $K$-dim one-hot label for node $v$. We adopt superscript $^L$ for labeled nodes (i.e. $\mathcal{V}^L$, $\mathbf{X}^L$, and $\mathbf{Y}^L$) and superscript $^U$ for the remaining unlabeled nodes (i.e. $\mathcal{V}^U$, $\mathbf{X}^U$, and $\mathbf{Y}^U$).

**Graph Neural Network.** Most GNNs follow the message-passing framework, where the representation $\mathbf{h}_v$ of node $v$ is updated by aggregating messages from its neighbors $\mathcal{N}_v$. For the $l$-th layer, $\mathbf{h}_v^l$ is obtained from the previous layer's representations of its neighbors as follows:

$$h_{N(v)}^{(l)} = \text{AGGR}(\{h_u^{l-1} : u \in \mathcal{N}_v\}) \tag{1}$$

$$h_v^{(l)} = \text{UPDATE}(h_{N(v)}^{(l)}, h_v^{l-1}), \tag{2}$$

where AGGR and UPDATE denote the aggregate and update operations, respectively.

**Transductive vs Inductive.** There are two settings for graph learning: transductive and inductive. In the former, models utilize all node features and graph edges. While the latter splits the unlabeled data into disjoint inductive subset and observed subset (i.e. $\mathcal{V}^U = \mathcal{V}_{obs}^U \cup \mathcal{V}_{ind}^U$ and $\mathcal{V}_{obs}^U \cap \mathcal{V}_{ind}^U = \emptyset$). Edges between $\mathcal{V}_{obs}^U$ and $\mathcal{V}_{ind}^U$ are preserved (Table 2).

Table 2: The inputs for GNNs and MLPs in different settings: transductive (*tran*) and inductive (*ind*). KD denotes the employed features for knowledge distillation. $\mathbf{H}$ denotes the graph nodes representations from GNN teachers.

| Model | Setting | Train | Test | KD |
|---|---|---|---|---|
| GNN | *tran* | $(\mathbf{X}, \mathcal{G}, \mathbf{Y}^L)$ | $(\mathbf{X}^U, \mathcal{G}, \mathbf{Y}^U)$ | $\mathbf{H}$ |
| | *ind* | $(\mathbf{X}^L, \mathcal{G}_{obs}, \mathbf{X}_{obs}^U, \mathbf{Y}^L)$ | $(\mathbf{X}_{ind}^U, \mathbf{Y}_{ind}^U)$ | $\mathbf{H}^L \cup \mathbf{H}_{obs}^U$ |
| MLP | | $(\mathbf{X}^L, \mathbf{Y}^L)$ | $(\mathbf{X}^U, \mathbf{Y}^U)$ | - |

## 4 METHODOLOGY

In this section, we first study the impact of graph structures (graph edges) on GNNs. Based on the analysis, we propose the PGKD to transfer the graph structural information from GNNs to MLPs.

### 4.1 IMPACT OF GRAPH STRUCTURES ON GNNS

**Intra-class Edges.** The propagation mechanisms in GNNs are the optimal solution for optimizing a feature fitting function with a graph Laplacian regularization term (Zhu et al., 2021; Ma et al., 2021). The Laplacian regularization guides the smoothness of $\mathbf{H}$ over $\mathcal{G}$, where connected nodes share similar features. Therefore, the homophily property for nodes from the same class can be captured. As shown in Table 5, the average connected node distance for GNNs is much smaller than that in initial node features.

**Inter-class Edges.** For inter-class edges, the nodes connected belong to different classes, known as heterophily. We define the class distance as the distance between class prototypes, and the class prototype is the prototypical vector for all nodes from the same class. The inter-class edges determine the pattern of class distances: for class $i, j, k$, if the inter-class edges between class $i, j$ are more than those between class $i, k$, then the class distance of $i, j$ would be smaller. From Table 6, we can infer that two classes would be closer if more inter-class edges existed between them in GNNs.

### 4.2 PROTOTYPE-GUIDED KNOWLEDGE DISTILLATION

Since the impact of graph structures (i.e. graph edges) on GNNs has been studied, the next goal is to distill such graph structural information from GNNs to MLPs. In PGKD, we design two losses to mimic the impact of intra-class and inter-class edges via class prototypes. Figure 1 shows an overview of PGKD.

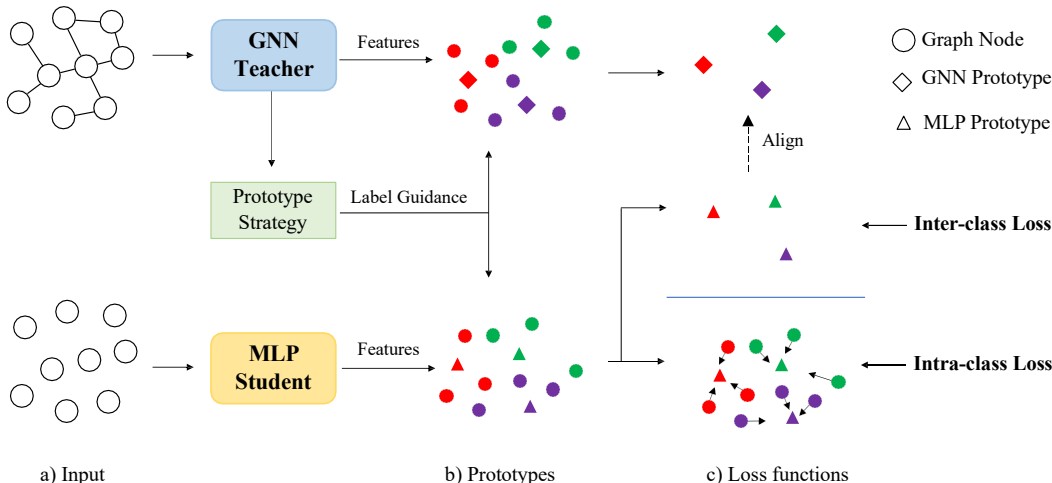

Figure 1: Overview of the proposed PGKD. The input is the whole graph for the GNN teacher but only the corresponding graph nodes for the MLP student. The circles mean the vectors for graph nodes and the same color denotes the same class. After getting the class prototypes, we design inter-class and intra-class loss to distill the graph structural information from the GNN teacher to the MLP student.

**Prototype Strategy.** To get the class prototype, the key is to label all nodes. In the inductive scenario, we employ the corresponding ground-truth label $\mathbf{Y}^L$ of training data for both GNN teachers and MLP students. However, in transductive scenarios, applying the ground-truth label $\mathbf{Y}$ would lead to label leaking. Hence, we employ the predicted label of the GNN teacher to label the MLP nodes. After grouping the nodes via the labels, we define the class prototypes as the mean vectors of all nodes from the same class. Henceforth, we use $(\mathbf{P}_1^t, ..., \mathbf{P}_K^t)$ and $(\mathbf{P}_1^s, ..., \mathbf{P}_K^s)$ to denote the GNN and MLP class prototypes, respectively.

**Intra-class Loss.** The intra-class edges in GNNs capture the homophily property for nodes from the same class. One intuitive idea is to draw closer for any two nodes from the same classes in the edge-free setting. However, this strategy has *two drawbacks*: 1) It is easily influenced by the outliers, viz. noisy points; and 2) A high complexity $O(n^2)$ where $n$ denotes the quantity of a given class. To tackle these, we design the intra-class loss analogous to InfoNCE (van den Oord et al., 2018), whose goal is to **draw a selected node closer to its corresponding prototype**. For node $i$ and its given label $c_i$, the loss is calculated as:

$$\mu_i = [d_1(h_i, \mathbf{P}_1^s), ..., d_1(h_i, \mathbf{P}_K^s)] \tag{3}$$

$$\mathcal{L}_{intra} = \Phi_1(\text{Softmax}(-\mu_i/\tau_1), c_i), \tag{4}$$

where $\Phi_1$ denotes loss functions such as cross-entropy loss, $\tau_1$ denotes the temperature parameter, $\mathbf{P}_1^s, ..., \mathbf{P}_K^s$ denote the MLP prototypes, and $d_1$ denotes the distance function. The class prototypes are less sensitive to noisy points. Meanwhile, the compute is decreased to $O(n \times K)$ where typically $K \ll n$.

**Inter-class Loss.** The inter-class edges determine the pattern of class distances. The class prototypes would be closer if more inter-class edges connect these two classes. However, we cannot count the edges in an edge-free setting. One solution is to force the student to **mimic the distance pattern of GNN teachers**. Aligning the distances directly cannot work, since the node representations from teacher and student lie in different semantic spaces. Therefore, we compute the relative distance in PGKD. The loss to align GNN prototype $\mathbf{P}_i^t$ and MLP prototype $\mathbf{P}_i^s$ is calculated by:

$$\sigma_i^t = [d_2(\mathbf{P}_i^t, \mathbf{P}_1^t), ..., d_2(\mathbf{P}_i^t, \mathbf{P}_K^t)] \tag{5}$$

$$\sigma_i^s = [d_2(\mathbf{P}_i^s, \mathbf{P}_1^s), ..., d_2(\mathbf{P}_i^s, \mathbf{P}_K^s)] \tag{6}$$

$$\mathcal{L}_{inter} = \Phi_2(\text{Softmax}(\sigma_i^s/\tau_2), \text{Softmax}(\sigma_i^t/\tau_2)), \tag{7}$$

where $\Phi_2$ denotes the similarity function for two distributions such as KL-divergence, $\tau_2$ denotes temperature parameter, $\mathbf{P}^t$ and $\mathbf{P}^s$ denote the prototypes, and $d_2$ denotes the distance function.

**Overall Target.** In PGKD, the overall loss function is:

$$\mathcal{L} = \mathcal{L}_{label} + \mathcal{L}_{kd} + \lambda_1 \mathcal{L}_{intra} + \lambda_2 \mathcal{L}_{inter}, \tag{8}$$

where $\mathcal{L}_{label}$ and $\mathcal{L}_{kd}$ are the original loss for classification and vanilla logit-base KD loss, respectively. $\lambda_1$ and $\lambda_2$ are hyperparameters. This way, the baseline GLNN is a *special case* of PGKD when both $\lambda_1$ and $\lambda_2$ are zeroed.

## 5 EXPERIMENTS

### 5.1 DATASETS

To evaluate the performance of PGKD, we consider seven popular benchmarks, including five homophilous graph datasets, namely, Cora (Sen et al., 2008), Citeseer (Sen et al., 2008), Pubmed (Namata et al., 2012), A-computer (Shchur et al., 2018), and Arxiv (Hu et al., 2020), and two heterophilous graph datasets, namely, Penn94 (Lim et al., 2021) and Twitch-gamer (Lim et al., 2021). In particular, Twitch-gamer and Arxiv are *large datasets* with more than 160,000 nodes. Please refer to Appendix A.1 for details of all datasets. We split these datasets for train/validation/test following GLNN (Zhang et al., 2022) for fair comparison. For the metric, we report the average accuracy on test data over five runs with different random seeds.

### 5.2 IMPLEMENTATION

**GNN Teacher.** To evaluate the ability on different backbones, we select four popular GNNs as the teacher model: GraphSAGE (Hamilton et al., 2017), GAT (Velickovic et al., 2017), GCN (Kipf & Welling, 2017) and APPNP (Klicpera et al., 2019), and perform experiments under both transductive and inductive settings.

**Baselines.** For baselines, we *do not* compare with the regularization methods since these methods utilize the graph edges as extra inputs. In real-world applications, these graph edges may be unavailable. Please refer to A.2 for specific scenarios. Therefore, we conduct all experiments in the *edge-free* setting. For fairness, we select the edge-free GLNN (Zhang et al., 2022) as our baseline, which adapts vanilla logit-base KD from GNNs to MLPs.

**Hyper-parameters.** We distill the two-layer GNN teacher to MLP student with two layers (on Cora, Citeseer, and A-computer) or three layers (on Pumbed, Penn94, Arxiv, and Twitch-gamer). For PGKD, we employ grid search to train the MLPs, where $\lambda_1$ is searched in $\{0.1, 0.2, 0.4\}$ and $\lambda_2$ in $\{0.05, 0.1\}$. We set $\tau_1$ and $\tau_2$ as 1 and 10, respectively. The hidden state dimension is 128 for both GNNs and MLPs. In all the datasets, the MLPs are trained for 500 epochs with early stopping.

### 5.3 MAIN RESULTS

We conduct experiments on seven benchmarks and select SAGE as GNN teachers for small datasets (Cora, Citeseer, and A-computer) and GCN for large datasets (Penn94, Pubmed, and Twitch-gamer). Meanwhile, we reproduce GLNN from its official codes. Table 3 reports the accuracy results. Several observations are in place:

- PKGD outperforms GLNN on all seven benchmarks with higher average scores under both transductive and inductive settings, thus demonstrating the effectiveness of PGKD in capturing graph structural information for the MLPs. In particular, PGKD achieves 76.35% on Pubmed under inductive setting, 1.86% higher than GLNN. PKGD can even outperform GNN teachers on some datasets (Citeseer and Pubmed).

- The standard deviations of PGKD are smaller than GLNN for almost all datasets, showing the stability and robustness of PGKD. For instance, PGKD gets 0.39% on A-computer under transductive setting, approximately $3\times$ smaller than the 1.04% of GLNN.

- Particularly, for *large dataset* (Arxiv and Twitch-gamer), PGKD statistically significantly outperforms GLNN. Specifically, the p-values for Arxiv and Twitch-gamer are 0.00/0.04 and 0.00/0.04 (transductive/inductive), respectively. Such results on the large datasets prove the effectiveness of PGKD.

Table 3: Experiment results of PGKD on several benchmarks under *transductive* and *inductive* settings. We report the average test accuracy (%) and the standard deviation over five runs for each dataset. PGKD outperforms GLNN with higher average scores and lower standard deviations. In particular, for large datasets (Arxiv and Twitch-gamer) with more than 160,000 nodes, PGKD statistically significantly (p-value < 0.05) outperforms GLNN.

| Dataset | GNN Setting | GNN | GLNN | PGKD (Ours) | $\Delta$GLNN |
|---|---|---|---|---|---|
| Cora | *transductive* | 81.11±2.05 | 80.22±1.81 | 82.15±0.19 | ↑ **1.93** |
|  | *inductive* | 81.59±1.95 | 74.38±0.85 | 74.85±0.24 | ↑ **0.47** |
| Citeseer | *transductive* | 70.62±1.53 | 71.87±1.90 | 72.93±1.17 | ↑ **1.06** |
|  | *inductive* | 69.89±2.80 | 69.34±2.10 | 69.94±2.03 | ↑ **0.60** |
| A-computer | *transductive* | 82.70±0.69 | 82.84±1.04 | 83.42±0.39 | ↑ **0.58** |
|  | *inductive* | 83.12±0.88 | 80.94±0.83 | 81.78±0.50 | ↑ **0.84** |
| Penn94 | *transductive* | 82.08±0.40 | 81.08±0.50 | 81.51±0.48 | ↑ **0.43** |
|  | *inductive* | 81.95±0.52 | 71.67±0.52 | 72.18±0.50 | ↑ **0.51** |
| Arxiv | *transductive* | 70.92±0.34 | 63.46±0.46 | 64.84±0.25 | ↑ **1.38** |
|  | *inductive* | 71.00±0.28 | 59.50±0.34 | 59.97±0.44 | ↑ **0.47** |
| Pubmed | *transductive* | 76.44±2.44 | 76.66±3.34 | 77.02±2.85 | ↑ **0.36** |
|  | *inductive* | 75.68±2.80 | 73.95±5.56 | 76.35±2.21 | ↑ **1.86** |
| Twitch-gamer | *transductive* | 62.58±0.19 | 60.07±0.16 | 60.56±0.07 | ↑ **0.49** |
|  | *inductive* | 62.34±0.44 | 59.57±0.30 | 60.01±0.42 | ↑ **0.44** |

Table 4: The ablation accuracy (%) on **Citeseer**, **Cora**, **Penn94**, and **Twitch-game** datasets under *transductive* setting. We report the average results for five runs.

| Dataset | Citeseer | Cora | Penn94 | Twitch-game |
|---|---|---|---|---|
| GNN | 71.85±1.40 | 82.24±0.59 | 82.08±0.40 | 62.58±0.19 |
| GLNN | 72.60±2.15 | 81.43±0.18 | 81.08±0.50 | 60.07±0.16 |
| PGKD | 73.20±1.58 | 82.39±0.64 | 81.51±0.48 | 60.56±0.07 |
| $-\mathcal{L}_{intra}$ | 72.17±1.55 | 81.12±0.92 | 81.18±0.17 | 60.11±0.23 |
| $-\mathcal{L}_{inter}$ | 72.15±1.68 | 81.91±0.54 | 81.53±0.34 | 60.54±0.08 |

## 5.4 ABLATION STUDIES

To better understand PGKD, we conduct ablation experiments on intra-class loss and inter-class loss. Without loss of generality, we select GCN as GNN teachers and compare the performance on Citeseer, Cora, Penn94, and Twitch-gamer.

Table 4 shows the experiment results, wherein the performance drops when either intra-class loss or inter-class loss is removed, indicating that both intra-class information and inter-class information are crucial. Moreover, it is interesting to see that PGKD with one loss exclusively would perform worse or slightly better than GLNN, but gains larger improvement than GLNN with two losses together. For example, PGKD gets 71.17% and 72.15% on Citeseer with one loss exclusively, both are lower than 72.60% of GLNN. However, PGKD would get a higher 73.20% than GLNN with both two losses. Such a phenomenon indicates that simultaneously considering both intra-class and inter-class information is crucial for effective MLP training.

## 6 ANALYSIS AND DISCUSSION

We further explore the ability to capture graph structural information as well as the robustness of the proposed PGKD. We also visualize the distributions of node representations for deeper insights.

## 6.1 CAN PGKD DISTILL THE IMPACT OF GRAPH EDGES?

As mentioned in Section 4.1, the intra-class edges guarantee the homophily for nodes from the same class, while the inter-class edges determine the pattern of distances among class prototypes.

We adopt SAGE as the GNN teacher and perform experiments under a transductive setting, and then calculate the average L2 distance for the features of connected nodes in the graph. Table 5 shows the average distance of initial node features and node features from GNN teacher (SAGE), GLNN, and PGKD. The distance of the GNN teacher is the shortest due to the information aggregation operations along

Table 5: Average L2 distance for the features of connected nodes on different datasets.

| Dataset | Input | GNN | GLNN | PGKD |
|---------|-------|------|------|------|
| Cora | 4.40 | 1.95 | 3.02 | 2.47 |
| Citeseer | 5.66 | 1.40 | 3.10 | 0.82 |
| A-computer | 17.63 | 2.35 | 7.14 | 4.76 |

graph edges. Meanwhile, the distance for GLNN is much longer due to the weak awareness of such graph structural information. PGKD gets shorter distances than GLNN, showing a great ability to capture intra-class graph structural information. In particular, PGKD gets a L2 distance of 0.82 on Citeseer, which is shorter than 3.10 from the GLNN.

The inter-class edges determine the pattern of distances among class prototypes. Specifically, the prototypes of two classes would be closer with more inter-edges connecting them in GNNs. We take statistics on the class distances (defined as L2 distances among class prototypes) and quantity of corresponding inter-class edges. For qualitative analysis, we calculate the Spearman correlation. From Table 6, the GNN teacher has a low Spearman correlation, whereas GLNN

Table 6: Spearman correlation $\rho$ between class distances and inter-class edges quantity. $\rho \to -1$ indicates more negatively correlated.

| Dataset | GNN | GLNN | PGKD |
|---------|-------|-------|-------|
| Cora | -0.94 | -0.88 | -0.92 |
| Citeseer | -0.71 | -0.62 | -0.67 |
| A-computer | -0.75 | -0.60 | -0.77 |

shows a relatively high value. Meanwhile, the proposed PGKD, thanks to the intra-class loss, can better capture the intra-class graph structural information and exhibits a much lower correlation.

To conclude, PGKD can distill the graph structural information from GNNs into MLPs better than GLNN in the edge-free setting.

## 6.2 IS PGKD ROBUST TO NOISY NODE FEATURES?

To analyze the robustness of PGKD to noise, we further evaluate its performance after adding Gaussian noise of different levels to initial node features $X$. Specifically, we replace $X$ with $(1 - \alpha)X + \alpha\epsilon$, where $\epsilon$ denotes the isotropic Gaussian noise independent of $X$, and $\alpha \in [0, 1]$ controls the noise level. A larger $\alpha$ means a stronger noise. Figure 2 shows the performance of GNN, GLNN, and PGKD under different noise levels. On both Cora and Pumbed, PGKD outperforms GLNN consistently as the noise level ranges from 0.1 to 0.9. Particularly, PGKD could get better results than GAT and APPNP on Pumbed with $\alpha = 0.9$. These show that PGKD is more robust than GLNN under noisy input node features due to its ability to capture graph structural information.

## 6.3 IMPACT OF INDUCTIVE SPLIT RATIO

To evaluate the ability for less observed data under inductive setting, we conduct the experiments under different split ratios, defined as the ratio $|\mathcal{V}_{ind}^U|/|\mathcal{V}^U|$. A larger split ratio means less observed unlabeled data during training and more inductive unlabeled data for test (cf. Section 3). As shown in Figure 3, the performance of the GNN teacher is not monotonically decreasing since the way to split graph (i.e. the edges to remove) is also vital as the number of nodes for training. PGKD outperforms GLNN and GNN under all split ratios. Also, the performance of PGKD is more stable than GLNN. This proves that PKGD, explicitly capturing the graph structural information, is robust and effective under different inductive split ratios.

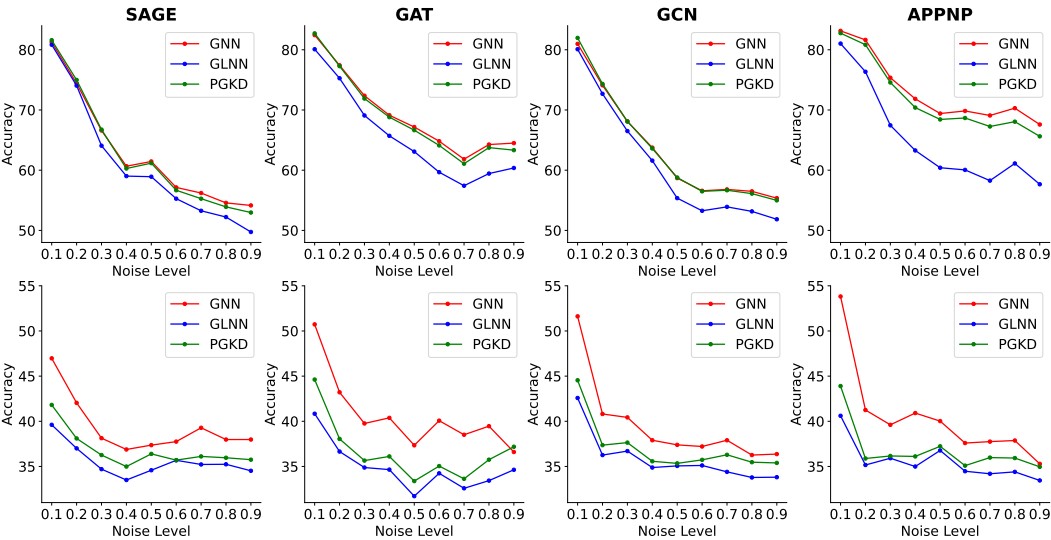

Figure 2: The performance of GNN teacher, distilled MLP students via GLNN and PKGD when adding different noise to the initial node features. For GNN teachers, we select SAGE, GAT, GCN and APPNP, respectively. **Upper**: **Cora** dataset and *transductive* setting. **Lower**: **Pubmed** dataset and *inductive* setting. when adding noise into node features, PGKD gets a little drop while GLNN drops a lot, showing the strong denoising ability of PGKD.

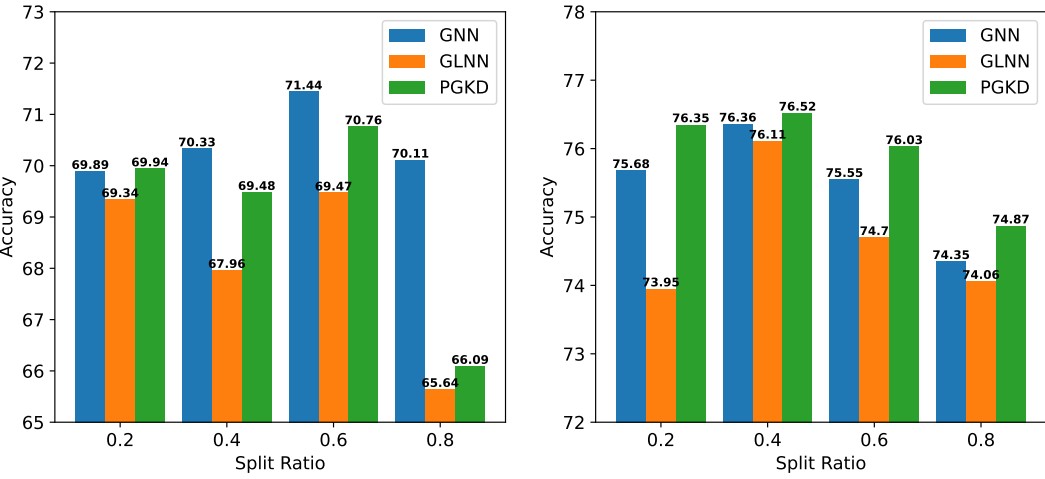

Figure 3: The performance of GNN teacher, distilled MLP students via GLNN and PKGD under *inductive* setting with different split ratios. **Left**: **Citeseer** dataset and SAGE as the GNN teacher. **Right**: **Pubmed** dataset and GCN as the GNN teacher.

### 6.4 IMPACT OF MLP SETTING

We further conduct experiments under different MLP settings. The GNN teacher is a two-layer GCN with 0.18M parameters and gets an accuracy of 83.37% on the Cora dataset. As shown in Table 7, the vanilla MLP shows an overfitting trend when the number of parameters increases, while PGKD does not. Meanwhile, PGKD gets the highest results under all settings and shows consistent improvement over GLNN. In particular, PGKD gets a score of 74.71% (#L=2, #H=128), which is 3.05% higher than GLNN. Such findings indicate that PGKD is more robust and effective in different MLP settings.

Table 7: Comparisons for vanilla MLP, distilled MLP students via GLNN and PGKD with different MLP settings on **Cora** under *inductive* setting. We report the average test accuracy (%). **#L** denotes the layers and **#H** denotes dimension of hidden state.

| #L | #H | Params | MLP | GLNN | PGKD | ∆GLNN |
|----|-----|--------|-------|-------|-------|-------|
| 2 | 64 | 0.09M | 53.40 | 73.30 | 74.00 | ↑**0.70** |
| 2 | 128 | 0.18M | 59.48 | 71.66 | 74.71 | ↑**3.05** |
| 3 | 128 | 0.20M | 54.33 | 73.07 | 74.24 | ↑**1.17** |
| 2 | 512 | 0.73M | 56.21 | 73.54 | 74.47 | ↑**0.92** |
| 3 | 512 | 1.00M | 54.57 | 72.83 | 74.00 | ↑**1.17** |

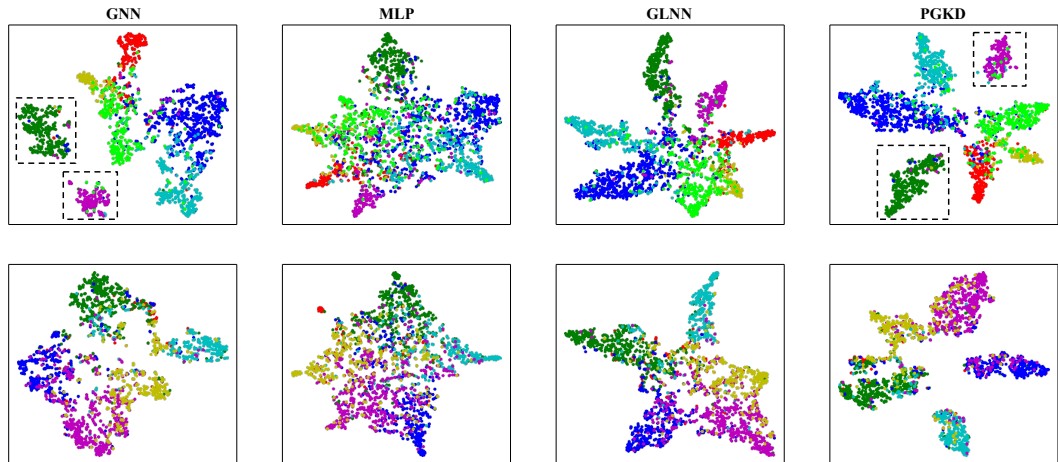

Figure 4: T-SNE visualization of the node representations for GNN teacher, vanilla MLP, and distilled MLPs from GLNN and PKGD. **Upper**: **Cora** dataset. **Lower**: **Citeseer** dataset.

## 6.5 NODE REPRESENTATION DISTRIBUTION

We visualize the distribution of node representations from GNNs and MLPs (vanilla MLPs without KD, MLPs from GLNN, and MLPs from PGKD) via t-SNE (van der Maaten & Hinton, 2008). We select GAT as the GNN teachers. Figure 4 shows the results on Cora and Citeseer under the transductive setting. Due to the message-passing architecture, the node representations in the same class from GNNs are much more gathered than vanilla MLPs. PGKD captures such graph information via intra-class loss, while vanilla MLPs and MLPs from GLNN lack such capability. The same-class features from both GLNN and vanilla MLP are slightly dispersed, while the features from PGKD are more clustered inside a class and separable between classes. Moreover, PGKD can learn class prototype distributions better. Specifically, in the GNN representations on Cora, the dark green and purple classes are far from each other. PGKD captures such a behavior well, where GLNN fails. The node distributions demonstrate the effectiveness of PGKD in distilling graph structural information.

## 7 CONCLUSION

A novel PGKD scheme has been proposed to distill the knowledge from high-accuracy GNNs to low-latency MLPs, wherein the distillation process is edge-free and the learned MLP students are structure-aware. Specifically, we analyze the impact of graph structure (graph edges) on GNNs and categorize them into intra-class and inter-class edges. Two corresponding losses via class prototypes are designed to transfer the graph structural knowledge from GNNs to MLPs. Experiments on popular benchmarks demonstrate the effectiveness of PGKD. Additionally, we show that PGKD is robust to noisy node features, and performs well under different training settings. For future work, PGKD will be generalized to other graph tasks beyond node classification. Another interesting direction will be to generate prototypes utilizing node representations rather than class labels.

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

# A  APPENDIX

## A.1  DATASETS

| Datasets | #Type | #Nodes | #Edges | #Features | #Classes |
|----------|-------|--------|--------|-----------|----------|
| Cora | Homo | 2,708 | 5,429 | 1,433 | 7 |
| Citeseer | Homo | 3,327 | 4,732 | 3,703 | 6 |
| A-computer | Homo | 7,650 | 119,081 | 745 | 8 |
| Pumbed | Homo | 19,717 | 44,324 | 500 | 3 |
| Arxiv | Homo | 169,343 | 1,166,243 | 128 | 40 |
| Penn94 | Heter | 41,554 | 1,362,229 | 5 | 2 |
| Twitch-gamer | Heter | 168,114 | 6,797,557 | 7 | 2 |

Table 8:  Statistics of the benchmarks. Homo and Heter denote homophilous and heterophilous graphs, respectively.

The details for datasets are shown in Table 8. In particular, Arxiv and Twitch-gamer are two large datasets with more than 160,000 nodes.

## A.2  NECESSITY FOR EDGE-FREE SETTING

For distillation from GNNs to MLPs, the edge-free setting means that edge information is not available for the distillation process.

**Goal**: the GNN teacher is trained by group $\mathcal{A}$, but the MLP student is from group $\mathcal{B}$. They need to distill the ability of the GNN teacher on one task to the MLP student.

**Edge-free setting**: the node features and corresponding GNN outputs are shared between group $\mathcal{A}$ and group $\mathcal{B}$ but the edge information is not shared.

The reasons are as follows:

- **privacy problem**: graph edges involve some privacy data and may be authorized for group $\mathcal{A}$ only, such as the edges from social relation graphs.

- **commercial consideration**: graph edges can be employed for other tasks, but group $\mathcal{A}$ wants to just share the ability of one task. For example, graph edges from custom-product graphs can be used for custom-custom social recommendations and product-product recommendations. When group $\mathcal{A}$ only wants to share the ability of custom-custom social recommendations, they would not share the edge information in case of ability leaking.

- **missing/corrupted edges**: GNN teacher was trained a long time before and graph edges are missing or corrupted.

## A.3  MORE STRATEGY FOR PROTOTYPES

In this paper, we select the prototypes as simple as the mean vectors. For a group of vectors $(V_1, V2, ..., V_n)$, the class prototype is defined as: $P := \frac{\sum V_1, V_2, ..., V_n}{n}$. For more comparison, we add an entropy-based approach to get prototypes. The entropy-based prototype is defined as: $P_{\text{entropy}} := \sum(w_1 V_1, w_2 V_2, ..., w_n V_n)$ and $(w_1, w_2, ..., w_n) := \text{Softmax}(\mathbb{E}(logit_1), \mathbb{E}(logit_2), ..., \mathbb{E}(logit_n))$, where $\mathbb{E}$ denotes the entropy function and $logit_i$ denotes the output logit of node $i$. The results are shown in Table 9.

Based on the results, we can see that the entropy-based prototype still outperforms GLNN but is slightly worse than the original prototype strategy, demonstrating the robustness of our method.

## A.4  HYPERPARAMETER SENSITIVITY ANALYSIS

There are four hyperparameters in PGKD, namely $\lambda_1$, $\lambda_2$, $\tau_1$, and $\tau_2$. We perform hyperparameter sensitive analyses on Citeseer (GraphSAGE as the teacher and under transductive setting).

Table 9: The comparison of different strategies to get prototypes.

| Dataset | GNN Setting | GNN | GLNN | PGKD | PGKD(Entropy) |
|---------|-------------|-----|------|------|---------------|
| Cora | *transductive* | 81.11±2.05 | 80.22±1.81 | 82.15±0.19 | 81.35±1.25 |
| | *inductive* | 81.59±1.95 | 74.38±0.85 | 74.85±0.24 | 74.52±1.22 |
| Citeseer | *transductive* | 70.62±1.53 | 71.87±1.90 | 72.93±1.17 | 72.12±1.28 |
| | *inductive* | 69.89±2.80 | 69.34±2.10 | 69.94±2.03 | 68.84±1.96 |

Table 10: Results of searching $\lambda_1$ and $\lambda_2$ when fixing $\tau_1 = 4$ and $\tau_2 = 10$.

| | $\lambda_1 = 0$ | $\lambda_1 = 0.05$ | $\lambda_1 = 0.1$ | $\lambda_1 = 0.2$ |
|---|---|---|---|---|
| $\lambda_2 = 0$ | 80.22±1.81 (GLNN) | 81.21±1.77 | 81.05±1.42 | 81.05±1.27 |
| $\lambda_2 = 0.01$ | 80.44±1.22 | 81.29±1.32 | 81.35±1.39 | 81.51±1.38 |
| $\lambda_2 = 0.05$ | 80.38±1.80 | 81.45±1.26 | 81.14±1.74 | 81.37±1.37 |

Table 11: Results of searching $\tau_1$ and $\tau_2$ when fixing $\lambda_1 = 0.1$ and $\lambda_2 = 0.01$.

| | $\tau_1 = 1$ | $\tau_1 = 2$ | $\tau_1 = 4$ | $\tau_1 = 10$ |
|---|---|---|---|---|
| $\tau_2 = 1$ | 81.29±1.46 | 81.57±1.25 | 81.27±1.36 | 80.89±1.28 |
| $\tau_2 = 2$ | 81.23±1.91 | 81.21±1.44 | 81.24±1.37 | 80.87±1.12 |
| $\tau_2 = 4$ | 81.41±1.51 | 81.31±1.46 | 81.10±1.32 | 80.85±1.28 |
| $\tau_2 = 10$ | 81.38±1.56 | 81.22±1.35 | 81.35±1.39 | 80.83±1.05 |

As shown in Table 10 and 11, We can find that PGKD always outperforms GLNN (80.22) as $\lambda_1$, $\lambda_2$, $\tau_1$, and $\tau_2$ change. Also, we can find that PGKD is more sensitive to $\tau_1$ and $\lambda_1$. Intuitively, the MLP students lack the GNN aggregation operations and thus the representations are more dispersed (also refer to Figure 4). The Intra-class loss would make the representations more clustered, which is beneficial for classification tasks.

## A.5 Comparison with GLNN

Compared to GLNN, PGKD is novel in terms of motivation, methodology, and contribution. In summary, the novelty of PGKD is threefold:

1. **Motivation**: GLNN is edge-free but **not** structure-aware. PGKD is both edge-free and structure-aware. This makes PGKD fundamentally different from GLNN, both in theory and in practice.

2. **Methodology**: We first analyze the impact of graph edges on GNNs. Then we design two novel losses based on our analyses.

3. **Contribution**: i) To the best of our knowledge, we are the **first** to study the impact of graph structures on GNNs by dividing the edges into intra-class edges and inter-class edges. The findings provide a deeper understanding of GNNs. ii) PGKD is the **first** method to distill GNN teachers to structure-aware MLP students under the edge-free setting. iii) We perform comprehensive experiments and ablation studies. The empirical results faithfully demonstrate the effectiveness and robustness of PGKD.

Moreover, GLNN can be viewed as *a special case* of proposed PGKD. The extra two losses help MLP students learn graph structural information from GNN teachers. Therefore, PGKD can get structure-aware MLPs in the edge-free setting.

