# OpenReview forum: "Edge-free but Structure-aware: Prototype-Guided Knowledge Distillation from GNNs to MLPs"
_ICLR.cc/2024/Conference — ICLR 2024 Conference Withdrawn Submission_

### Official Review · Reviewer_Yv3j · 2023-10-31

**Soundness:** 2 fair
**Presentation:** 2 fair
**Contribution:** 2 fair
**Rating:** 3
**Confidence:** 4

**Summary:**

The paper introduces Prototype-Guided Knowledge Distillation (PGKD), a technique for distilling Graph Neural Networks (GNNs) into low-latency, structure-aware Multilayer Perceptrons (MLPs) without requiring graph edge information. Utilizing class prototypes, the authors develop both intra-class and inter-class losses to facilitate this distillation process.

**Strengths:**

1. The manuscript is articulately composed and presents its content in a clear and coherent manner, making it accessible and straightforward for readers to follow. The overall quality of presentation within the paper is commendable.
2. The authors have dedicated a substantial portion of the paper (Section 6) to empirical analysis and detailed discussions of the method, providing valuable insights and a deeper understanding of the approach and its implications.

**Weaknesses:**

1. concerns about the effectiveness of PGKD:
    1) A crucial piece of information that is missing from the paper is the comparative performance of GLNN and a standard MLP. This comparison is vital as it establishes a baseline to understand the necessity of using PGKD. For instance, Lim et al. have demonstrated that a vanilla MLP can achieve a performance of 60.92 on the twitch-gamers dataset [1], which is slightly higher than what PGKD manages to achieve.

    2) The enhancement in performance that PGKD provides over GLNN is minimal, with the majority of improvements being less than 1%. This marginal gain puts into question the practical significance and impact of adopting the PGKD method.

2. Missing important related works:
    1) The authors claim that they are the first to study the impact of intra-class edges and inter class edges. However, there are a moderate number of existing studies in the literature, e.g,in [2].

    2) Missing a few related works for prototypes and graph neural networks, e.g, [3][4]

    3) The intra-class loss serves to attract a selected node towards its associated prototype, and while it is not identical, its objective aligns with that of the compatibility loss proposed in [3].


3. The authors emphasize latency reduction during inference as a key motivation for their work. Nevertheless, the paper lacks a crucial comparison of runtime between PKGD, GLNN, MLP, and GNN. Including such a comparison is vital to substantiate the authors' claims and showcase the practical benefits of PKGD in real-world applications.


References:

[1] Lim et al. "Large scale learning on non-homophilous graphs: New benchmarks and strong simple methods." Advances in Neural Information Processing Systems 34 (2021): 20887-20902.

[2] Chen et al. "Measuring and relieving the over-smoothing problem for graph neural networks from the topological view." Proceedings of the AAAI conference on artificial intelligence. Vol. 34. No. 04. 2020.

[3] Dong et al. "ProtoGNN: Prototype-Assisted Message Passing Framework for Non-Homophilous Graphs." (2022).

[4] Zhang et al. "Protgnn: Towards self-explaining graph neural networks." Proceedings of the AAAI Conference on Artificial Intelligence. Vol. 36. No. 8. 2022.

**Questions:**

1. What is the runtime comparison between PKGD, GLNN, MLP, and GNN?
2. What is the performance comparision between PKGD and MLP?
3. Regarding the motivation for edge-free setting (section A.2), could the authors provide more concrete examples or real-world scenarios where the edge information is strictly confidential or proprietary, yet the GNN outputs and node embeddings can be freely shared?

---

> ### Author Response · Authors · 2023-11-15
>
> Thanks for your reviews. For your concerns, our responses are as follows:
>
> **R4-Q1**:  Concerns about the effectiveness of PGKD.
>
> In this paper, we study the GNN2MLP distillation based on an edge-free setting. In the paper "Large scale learning on non-homophilous graphs: New benchmarks and strong simple methods", both LINK and LINKX employ the adjacent matrix as input which is not edge-free.
>
> For effectiveness, PGKD outperforms GLNN with higher average scores and lower standard deviations. In particular, for large datasets (Arxiv and Twitch-gamer) with more than 160,000 nodes, PGKD statistically significantly (p-value < 0.05) outperforms GLNN. Also, as shown in Figure 2, PGKD shows strong robustness to noisy features.
>
> **R4-Q2**:  Missing important related works.
>
> Thanks again for your kindness. We will add these papers to the next version.
>
> **R4-Q3**: The authors emphasize latency reduction during inference as a key motivation for their work. Nevertheless, the paper lacks a crucial comparison of runtime between PKGD, GLNN, MLP, and GNN. Including such a comparison is vital to substantiate the authors' claims and showcase the practical benefits of PKGD in real-world applications.
>
> For PKGD, GLNN, and MLP methods, the models we get and deploy are all MLPs of the same size. For PGKD, we do not add any parameters compared to GLNN. For the latency of the trained student model, GNN > GLNN==PGKD==MLP.

---

> ### Comment · Reviewer_Yv3j · 2023-11-15
>
> It appears there has been a misunderstanding regarding my question.
> "A crucial piece of information that is missing from the paper is the comparative performance of PGKD and a standard MLP. This comparison is vital as it establishes a baseline to understand the necessity of using PGKD. For instance, Lim et al. have demonstrated that a vanilla MLP can achieve a performance of 60.92 on the twitch-gamers dataset [1], which is slightly higher than what PGKD manages to achieve."
>
> My inquiry was aimed at comparing the performance of PGKD with that of a standard MLP, not PGKD versus LINKX. The figure 60.92% pertains to the MLP's performance as documented in [1], which surpasses that of the PGKD. The performance of the MLP serves as a critical benchmark. Despite your previous note that PKGD achieves statistically significant improvements over GLNN for the larger Twitch-gamer dataset, it nonetheless falls short of the results obtained with a basic MLP. This raises questions about the actual efficacy of PKGD.
>
> I would strongly urge the authors to include a comparison with a vanilla MLP as a baseline within the performance evaluation. Without this, justifying the use of PKGD over simpler models becomes challenging.

---

### Official Review · Reviewer_wb8Q · 2023-10-31

**Soundness:** 2 fair
**Presentation:** 2 fair
**Contribution:** 2 fair
**Rating:** 3
**Confidence:** 4

**Summary:**

This paper studies the problem of knowledge distillation from GNNs to MLPs, with the absence of edges.
The authors try to tackle this problem by introducing prototypes, which is a common strategy in KD and GNNs.
The authors conduct some experiments on some small datasets and compare with a few baselines, which can somehow prove the effectiveness of the proposed method.

**Strengths:**

* The problem of distillation from GNNs to MLP is interesting and useful.
* The paper is overall easy to follow.

**Weaknesses:**

* The motivation of introducing the prototype is not clear, it does not seems to tackle the problem of "structure-aware distillation". Although it can somehow satisfy the edge-free requirement, the advantage in distilling the structure information is not clear, which is more critical in GNN distillation.
* The so-called "the impact of graph structures (i.e. graph edges) on GNNs has been studied" is over-claimed. First, the conclusion based on the inter and intra edges is trival. Second, the analysis and conclusion are based on some small datasets, which is not sufficient.
* The overall model only introduce the prototypes for more information distillation, rather than overcoming the critical "structure-aware distillation" problem
* The evaluated dataset is too small, where the distillation is not necessary.
* The baselines are weak and many important baselines are missed.

**Questions:**

* How can the model distill the structure information?
* What's the advantage of introducing prototypes?
* Please provide more results on large scale datasets and more baselines.

---

> ### Author Response · Authors · 2023-11-15
>
> Thanks for your reviews. For your concerns, our responses are as follows:
>
> **R3-Q1**: The motivation for introducing the prototype is not clear, it does not seem to tackle the problem of "structure-aware distillation". Although it can somehow satisfy the edge-free requirement, the advantage in distilling the structural information is not clear, which is more critical in GNN distillation.
>
> Why choosing prototypes is based on the impact of edges we study. As explained in Section 4.1, the intra-class edges help capture the homophily property for nodes from the same class, and the inter-class edges determine the pattern of class distances. The class distances are defined as the distances between class prototypes. Also, the homophily property can be better mimicked by prototypes (Please refer to sec 4.2 for pros of prototypes). Therefore, we chose the prototype to design the losses.
>
> **R3-Q2**:  The so-called "the impact of graph structures (i.e. graph edges) on GNNs has been studied" is over-claimed. First, the conclusion based on the inter and intra edges is trivial. Second, the analysis and conclusion are based on some small datasets, which is not sufficient.
>
> The conclusion for inter-class edge is non-trivial.
> We analyze the results on cora, citeseer, and A-computer. However, the A-computer dataset contains 7,650 nodes and 119,081 edges with 745-dimensional features.
>
> **R3-Q3**:  The overall model only introduces prototypes for more information distillation, rather than overcoming the critical "structure-aware distillation" problem.
>
> Structure-aware means whether the MLP students learn the graph structural information. We analyze the graph structural information and then try to distill such information to MLP students without edges. Therefore, our PGKD method achieves the goal of learning structure-aware MLPs.
>
> **R3-Q4**:  The evaluated dataset is too small, where the distillation is not necessary.
>
> We evaluate our dataset on 7 datasets and the size can be found in Appendix A.1, including Twitvh-game (168114 nodes and 6797557 edges) and Arxiv (169343 nodes and 1166243 edges). Both are large datasets.
>
> **R3-Q5**:  The baselines are weak and many important baselines are missed.
>
> We have discussed many related methods in the related work section.
> As mentioned in section 5.2,
> > For baselines, we do not compare with the regularization methods since these methods utilize the graph edges as extra inputs. In real-world applications, these graph edges may be unavailable. Please refer to Appendix A.2 for specific scenarios. Therefore, we conduct all experiments in the edge-free setting. For fairness, we select the edge-free GLNN (Zhang et al., 2022) as our baseline, which adapts vanilla logit-based KD from GNNs to MLPs.
>
> As far as we know, the only edge-free baseline till now is GLNN.
>
> **R3-Q6**:  How can the model distill the structure information?
>
> Please refer to Section 6.1. We show the empirical results in Table 5 and Table 6. Also, we show the node distributions in Section 6.5 for visualization.
>
> **R3-Q7**:  What's the advantage of introducing prototypes?
>
> Please refer to Section 4.2 for more details.
> > The intra-class edges in GNNs capture the homophily property for nodes from the same class. One intuitive idea is to draw closer for any two nodes from the same classes in the edge-free setting. However, this strategy has two drawbacks: 1) It is easily influenced by the outliers, viz. noisy points; and 2) A high complexity O(n^2 ) where n denotes the quantity of a given class. To tackle these, we design the intra-class loss analogous to InfoNCE (van den Oord et al., 2018), whose goal is to draw a selected node closer to its corresponding prototype.

---

### Official Review · Reviewer_74TD · 2023-10-31

**Soundness:** 2 fair
**Presentation:** 3 good
**Contribution:** 2 fair
**Rating:** 5
**Confidence:** 4

**Summary:**

The paper introduces two regularization terms that help to control the process of knowledge distillation from a GNN teacher to MLP. The authors separate the impact of topology structures into two types, inter-class effects and intra-class effects, and encourage MLP to imitate the class embedding correlations. The experimental results demonstrate the effectiveness of this approach.

**Strengths:**

S1: The paper provides a novel aspect of the analysis of the effect of edges from inter-class effect and intra-class effect.

S2: The paper is well-motivated and clear.

S3: The method is simple but effective.

**Weaknesses:**

W1: A more comprehensive analysis is needed to explore the inter-class effects. While inter-class edges may impact the inter-class distribution, it's unclear whether mimicking GNN embeddings consistently yields the best classification results. An alternative approach, the supervised InfoNCE loss, compresses intra-class distribution and promotes uniform distribution of class prototypes in the embedding space for better class separability. To better understand the merits of PGKD, a comparison and analysis with supervised InfoNCE loss is warranted.

W2: When conducting experiments related to #H, it would be beneficial to include larger datasets. As indicated by GLNN, #H notably influences inductive results in larger datasets. It is advisable to evaluate PGKD in this context and provide a comparison to demonstrate its optimal performance, rather than solely assessing its performance on small datasets.

W3: Could you clarify the rationale behind selecting SAGE and GCN for small and large datasets? Are there specific reasons for this choice?

W4: While the hyperparameters \lambda 1 and \lambda 2 play a pivotal role in obtaining optimal results, the current search space for these parameters is rather constrained. What might be the impact of expanding this search space to encompass a broader range of values?

W5: Why does the inductive test of GNN not need the graph structure according to Table 2?

**Questions:**

Refer to weakness

---

> ### Author Response · Authors · 2023-11-15
>
> Thanks for your reviews. We are glad to see that you find our PGKD provides a novel aspect of the analysis of the effect of edges from inter-class effect and intra-class effect. And our paper is well-motivated and clear. For your concerns, our responses are as follows:
>
> **R2-Q1**:   Could you clarify the rationale behind selecting SAGE and GCN for small and large datasets? Are there specific reasons for this choice?
>
> We just select them randomly to test the generality of different GNN teachers. In Figure 2, we show the performance of SAGE, GAT, GCN, and APPNP for more results.
>
> **R2-Q2**:  While the hyperparameters \lambda 1 and \lambda 2 play a pivotal role in obtaining optimal results, the current search space for these parameters is rather constrained. What might be the impact of expanding this search space to encompass a broader range of values?
>
> Thanks for your suggestions. We do believe that searching hyperparameters in larger space could lead to better results. However, for larger datasets such as Arxiv and Twitch-gamer, the search cost is too high to afford.
> For more results, please refer to Table 10 and Table 11 in the Appendix.
>
> **R2-Q3**:  Why does the inductive test of GNN not need the graph structure according to Table 2?
>
> Under the inductive setting, the goal is to test the general ability of unseen graph nodes since we often meet with unseen nodes in realistic scenarios.
>
> **R2-S1**:  Other suggestions.
>
> We sincerely thank you for your insightful suggestions. We would consider adding them to our next version.

---

> > ### Comment · Reviewer_74TD · 2023-11-22
> >
> > Thanks for your response. As some main problems (e.g., W1 and W2) have not been addressed, I keep my initial score of 5.

---

### Official Review · Reviewer_x8qx · 2023-11-01

**Soundness:** 3 good
**Presentation:** 3 good
**Contribution:** 2 fair
**Rating:** 3
**Confidence:** 5

**Summary:**

The paper presents a method called Prototype-Guided Knowledge Distillation (PGKD) to distill high-accuracy Graph Neural Networks (GNNs) into low-latency Multilayer Perceptrons (MLPs) for graph tasks. Unlike conventional approaches, PGKD does not require graph edges but effectively captures graph structural information by distilling it from GNNs. The authors propose two prototype-based losses to facilitate this knowledge transfer. Experimental results on popular graph benchmarks demonstrate the effectiveness and robustness of PGKD.

**Strengths:**

The proposed Prototype-Guided Knowledge Distillation (PGKD) presents a novel approach to address the limitation of conventional MLPs in capturing graph structural information, especially in an edge-free setting. This problem formulation is interesting and brings forth a creative combination of knowledge distillation and prototype learning.

The paper has conducted experiments on popular graph benchmarks, which adds some credibility to the empirical evaluation of the proposed method. The integration of prototype learning with knowledge distillation to distill graph structural information from GNNs to MLPs shows a thoughtful attempt to improve model interpretability and performance.

**Weaknesses:**

1. I don't fully understand why the distilled model with prototypical label can be called "structure-aware". It looks like another supervision we got from GNN over original graph. Given this definition, the original GLNN can also be regarded structure-aware. Can the authors explain this part more clearly, besides the prototypical label, what's additional difference?

2. Seems the only compared baseline is GLNN. Can the authors compare with some other MLP baselines (e.g., Graph Random Network)

This paper discusses a few such baselines: https://openreview.net/forum?id=tiqI7w64JG2

**Questions:**

I'm still not fully convinced why a prototypical cluster can help distillation, is it only work for graph or can be generalized to other datasets? Can the authors provide some explanation?

How is this work differ from some prototypical contrastive learning: https://openreview.net/pdf?id=KmykpuSrjcq and https://arxiv.org/abs/2012.12533

---

> ### Author Response · Authors · 2023-11-15
>
> Thanks for your reviews. We are glad to see that you find our problem formulation interesting and proposed PGKD brings forth a creative combination of knowledge distillation and prototype learning. For your concerns, our responses are as follows:
>
>
> **R1-Q1**:   Why PGKD is structure-aware but GLNN is not.
>
> As mentioned in Section 5.8 of the GLNN paper,
> > For example, when every node is labeled by its degree or whether it forms a triangle. Then MLPs won’t be able to fit meaningful functions, and neither will GLNNs. We can see that GLNN only learns from the node distributions of GNN and ignores the edge information, and thus we consider GLNN as not structure-aware.
>
> For PGKD, the MLP students do not only learn the node distributions from GNN teachers but also the impact of inter-class edges and intra-class edges.
>
> **R1-Q2**:   Why compare to GLNN only.
>
> As mentioned in section 5.2,
> > For baselines, we do not compare with the regularization methods since these methods utilize the graph edges as extra inputs. In real-world applications, these graph edges may be unavailable. Please refer to Appendix A.2 for specific scenarios. Therefore, we conduct all experiments in the edge-free setting. For fairness, we select the edge-free GLNN (Zhang et al., 2022) as our baseline, which adapts vanilla logit-based KD from GNNs to MLPs.
>
> As far as we know, the only edge-free baseline till now is GLNN.
>
> **R1-Q3**:   Why prototypical cluster helps distillation? is it only work for graphs or can it be generalized to other datasets? Can the authors provide some explanation? How is this work different from some prototypical contrastive learning: https://openreview.net/pdf?id=KmykpuSrjcq and https://arxiv.org/abs/2012.12533
>
> Why choosing prototypes is based on the impact of edges we study. As explained in Section 4.1, the intra-class edges help capture the homophily property for nodes from the same class, and the inter-class edges determine the pattern of class distances. The class distances are defined as the distances between class prototypes. Also, the homophily property can be better mimicked by prototypes (Please refer to sec 4.2 for pros of prototypes). Therefore, we chose the prototype to design the losses.
>
> PGKD works for the situation aiming to distill the knowledge from GNNs to struct-aware MLPs under the edge-free setting. For other scenarios, it may work if the motivation and setting are similar.
>
> Differences:
>
> i) Compared to PCL, the overall idea is similar and they design the fine-grained and coarse-grained prototypes. For PGKD, we only need to calculate one prototype for each class. Both methods employ the idea of contrastive learning. We would consider adding it to the Related Work section.
>
> ii) MICRO-Graph is graph level but our PGKD is node level. Also, the prototypes in PGKD are related to node labels but the MICRO-Graph is self-supervised.